# GROUNDED VIDEO CAPTION GENERATION

## ABSTRACT

We propose a new task, dataset and model for grounded video caption generation. This task unifies captioning and object grounding in video, where the objects in the caption are grounded in the video via temporally consistet bounding boxes. We introduce the following contributions. First, we present a task definition and a manually annotated test dataset for this task, referred to as GROunded Video Caption Generation (GROC). Second, we introduce a large-scale automatic annotation method leveraging an existing model for grounded still image captioning together with an LLM for summarising frame-level captions into temporally consistent captions in video. Furthermore, we prompt the LLM to track by language – classifying noun phrases from the frame-level captions into noun phrases of the video-level generated caption. We apply this approach to videos from the HowTo100M dataset, which results in a new large-scale training dataset, called HowToGround, with automatically annotated captions and spatio-temporally consistent bounding boxes with coherent natural language labels. Third, we introduce a new grounded video caption generation model, called VideoGLaMM, and train the model on the new automatically annotated HowToGround dataset. Finally, results of our VideoGLaMM model set the state of the art for the new task of grounded video caption generation. We perform extensive ablations and demonstrate the importance of key technical contributions of our model.

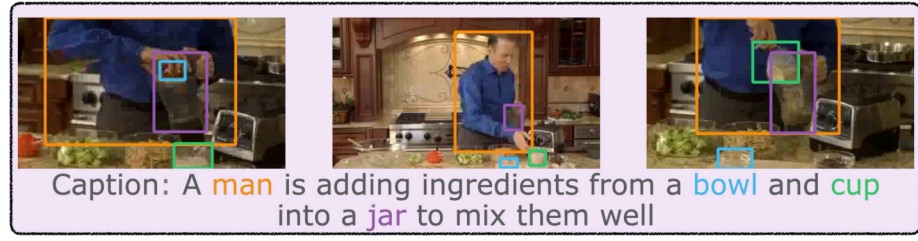

Figure 1: **The GROunded video Caption generation task.** Three frames from an example video from our new manually annotated GROC dataset of natural language descriptions grounded with temporally consistent bounding boxes in videos.

## 1 INTRODUCTION

In recent years, we have witnessed tremendous progress in multimodal video understanding thanks to Large Language Models and advanced designs that exploit the synergy of video and language. Current efforts have focused on a variety of tasks that require reasoning about (possibly long) videos together with a certain level of comprehension of natural language. Examples include natural language video captioning (Chen & Jiang, 2021; Fujita et al., 2020; Huang et al., 2020; Mun et al., 2019; Tang et al., 2021a; Wang et al., 2018; 2021; Zhou et al., 2018), temporal alignment of video with language (Han et al., 2022; Ko et al., 2022; Sigurdsson

et al., 2020; Yang et al., 2021b), finding relevant moments in videos given a language query (Gao et al., 2017; Hendricks et al., 2018; Lei et al., 2020b; Zhang et al., 2020b;a; Zhu et al., 2022), or video question answering (Engin et al., 2021; Kim et al., 2020; Le et al., 2020; Lei et al., 2018; Li et al., 2019; Park et al., 2021; Yang et al., 2021a; Yu et al., 2018; Yang et al., 2022a; Zeng et al., 2017). Others have looked at producing bounding boxes of events in video given natural language descriptions (Tang et al., 2021b; Zhang et al., 2020d; Huang et al., 2018) or given natural language questions (Lei et al., 2020a). Overall, these efforts have focused on producing video-level or moment-level outputs, such as (temporally localized) captions, or producing single event-level bounding boxes in video.

At the same time, producing natural language video descriptions where the described objects are spatio-temporally grounded with bounding boxes in videos has received much less attention. Progress on this problem is, however, important as spatio-temporal grounding of natural language on a large scale is an important step to advance areas such as human-robot interaction and embodied perception (Li et al., 2022; McCarthy et al., 2024; Patel et al., 2022; Sermanet et al., 2018; Zorina et al., 2021). Key factors limiting progress on this problem are the lack of annotated testing data, dedicated grounded video caption generation models and appropriate large-scale training datasets, which are costly to manually annotate.

In this work, we aim to narrow this gap by the following four contributions. **First**, we introduce the grounded video caption generation task together with a manually annotated test dataset of 1000 videos, which we name GROunded Video Caption Generation (GROC). This allows to measure progress on this challenging problem. **Second**, to address the issue of limited training data, we introduce a large-scale automatic annotation method leveraging an existing model for grounded still image captioning together with an LLM to summarize frame-level captions into video-level captions. The LLM is also tasked to *track by language*, associating frame-level phrases that correspond to objects with video-level phrases, resulting in object tubes with a consistent label. We apply this approach to videos from the HowTo100M dataset, which results in a new large-scale training dataset, called HowToGround, with automatically annotated captions and spatio-temporally consistent bounding boxes with coherent natural language labels. **Third**, we introduce a new grounded video caption generation model, called VideoGLaMM. The key technical contributions of this model include: (i) spatio-temporal adapters, which enable efficient modeling of spatio-temporal information in video; (ii) bounding box decoder and that outputs temporally coherent bounding boxes in video and (iii) temporal objectness head that explicitly models objects that temporally leave the frame or are occluded. We train the VideoGLaMM model on the automatically annotated HowToGround dataset. **Fourth**, we perform extensive ablations and demonstrate the importance of key technical contributions of our model. Results of our VideoGLaMM model set the state of the art for the new task of grounded video caption generation.

## 2 RELATED WORK

**Image-based grounded data generation**. Recently, there has been an increased interest in developing large multi-modal models capable of grounding text to images as well as comprehending referring expressions (Zhang et al., 2023; Peng et al., 2023; Zhao et al., 2023; Chen et al., 2023; Ma et al., 2024). The scale of these models dictates that they should be trained on large-scale datasets. As obtaining grounding datasets manually is extremely laborious, particularly at this scale, methods typically resort to pretrained models to harness these data. Zhang *et al.* introduced the grounded visual chat task and generated a dataset for the task using GPT-4 (OpenAI et al., 2024) along with visual instruction tuning data that are paired with ground truth bounding boxes. Rasheed et al. (2024) introduced a complex automated annotation pipeline for grounded conversation generation consisting of multiple stages and resorting to a variety of pretrained models to obtain a large scale pseudolabelled dataset. Peng et al. (2023) proposed to use Part-of-Speech tagging for extracting noun chunks from the captions and fed them to a pretrained grounding module for pairing them with bounding boxes. Chen et al. (2023) proposed a model for referential dialogue. To train the model, they combined existing grounding and referring expression comprehension data along with QA pairs associated with bounding boxes generated with GPT-4. We build on this line of work but focus on the video domain.

**Datasets for spatio-temporal grounding in video**. The existing datasets (Tang et al., 2021b; Zhang et al., 2020d; Huang et al., 2018) for spatio-temporal video grounding are relatively small scale as they rely on manual annotation, which is tedious and time consuming. Typically, the existing datasets also focus on a single grounding bounding box (Tang et al., 2021b; Zhang et al., 2020d), which can be a limiting factor in instructional videos where multiple objects are often manipulated. Finally, the focus is often on the task of predicting bounding boxes given a natural language description (Huang et al., 2018) or natural language questions in a video question answering task (Lei et al., 2020a). In contrast, we create an automatic procedure for pseudo annotation of spatio-temporally grounded captions, which allows us to create a large-scale dataset. In addition, we focus on the task of grounded video caption generation, which requires generating both the natural language description as well as multiple bounding boxes grounding individual described objects in the video.

**Methods for spatio-temporal grounding in video**. Spatio-temporal video grounding (Tang et al., 2021b; Zhang et al., 2020d; Yang et al., 2022b; Tan et al., 2021; Lin et al., 2023; Chen et al., 2024; Su et al., 2021; Zhang et al., 2020c; Jin et al., 2022; Gu et al., 2024; Wasim et al., 2024) is the task where given a natural language description and a video, a model should predict a single spatio-temporal tube enclosing the full event described in the text. Zhang et al. (2020d) introduce a spatio-temporal graph encoder for multi-step reasoning using as input features of the detected objects. Many works have since relied on object detection features for spatio-temporal grounding. Tang et al. (2021b) extract object proposals and feed both language tokens and detection features into a multi-modal transformer. Yang et al. (2022b) adapt the MDETR architecture to spatio-temporal video grounding. Tan et al. (2021); Chen et al. (2024) propose a multi-modal contrastive learning frame-work for spatio-temporal grounding by training on videos from HowTo100M. Lin et al. (2023) introduce a two-stream architecture for modelling appearance and motion. Gu et al. (2024) introduced a Context-Guided Decoder where queries are enhanced with rich visual cues generated from an Instance Context Generation module. All these works are limited in that they do not have a module for text generation as in the spatio-temporal grounding task the text is given, nor can they generate multiple spatio-temporal tubes for multiple objects in the video. Our task formulation, automatic annotation method and the proposed model address these limitations.

## 3 GROunded Video Vaption Generation: task, datasets and metrics

**Task definition.** We introduce the **GRO**unded Video **C**aption Generation (GROC) task for video understanding. Similar to Grounded Caption Generation for images, the task is to both generate a caption for the video and also predict bounding boxes across frames for noun phrases from the sentence. Differently than the image-based task, where bounding boxes are always predicted for the tagged/predicted noun phrases from the sentence, in the GROC task, as objects might disappear in some frames, there should not be bounding boxes predictions for the disappearing objects in those frames. Therefore, the task has the extra difficulty that a mechanism is needed to decide at each frame of the video whether to predict a bounding box for a given phrase from the sentence or not. Additionally, the temporal dimension of videos adds extra challenges as bounding boxes need to be spatio-temporally smooth across frames. To summarise, given a video the GROC task entails: i) predicting a caption describing the video, ii) tagging noun phrases in the caption that correspond to objects that will be grounded, iii) predict bounding boxes across frames for the tagged objects from the caption. As already discussed, as objects can disappear and reappear, there might be discontinuities in the bounding box predictions; in other words, there can be more than one spatio-temporal tubes for each object with gaps in between that correspond to objects disappearing at certain frames due to occlusions.

**GROC: Manually annotated evaluation dataset for grounded video caption generation**. We manually annotate the GROC evaluation dataset using videos from the HowTo100M dataset (Miech et al., 2019). We provide manual annotations for 1100 video clips. We construct the validation set by randomly sampling 100 manually annotated clips. We form the test set using the remaining 1000 manually annotated clips. The annotation procedure consists of 2 steps: i) video selection where 'interesting' videos are selected from a

large pool of videos randomly drawn from HowTo100M, and ii) video annotation where videos are annotated with a caption, bounding boxes and noun phrases from the caption associated with the bounding boxes. The video annotation itself consists of 3 steps. The first step entails watching the video and providing a description of what is happening in the video and the objects that are being used. Note, that we are interested in the *active objects*, *i.e.* objects that humans interact with, rather than densely describing all objects in the scene. In the second step, we annotate with bounding boxes all visible instances of humans/objects from the caption that has been provided in the previous step. The bounding boxes are applied to all frames where the humans/objects are visible. Finally, we annotate each bounding box with a short phrase or word which should match exactly a short phrase/word from the caption. For more information about the annotation procedure including the exact annotation guidelines and the mechanisms for assuring the consistency and the overall quality of the annotations, please see Section F in the Appendix.

**Evaluation metrics**. We build on the metrics for grounding captions in still images Rasheed et al. (2024) and adapt them to our task. These include METEOR (Banerjee & Lavie, 2005) and CIDEr (Vedantam et al., 2015) for the quality of the captions, AP50 for the accuracy of grounding bounding boxes to phrases, mean IoU across videos to measure the bounding box detection quality, and the recall Rasheed et al. (2024) that combines IoU and the similarity of embeddings of the predicted phrases that correspond to bounding boxes. The aim of the recall metric is to assess the rate of positive predictions from the model, where a prediction is considered positive if both the IoU and the phrase similarity is above a certain threshold. We propose two different settings of AP50, mIoU and recall for the GROC task: i) frame-level evaluation where the metrics are calculated across all frames of all videos – same as for images and ii) video-level evaluation where the metrics are calculated per video and averaged across videos.

## 4 AUTOMATIC ANNOTATION METHOD AND HOWTOGROUND DATASET

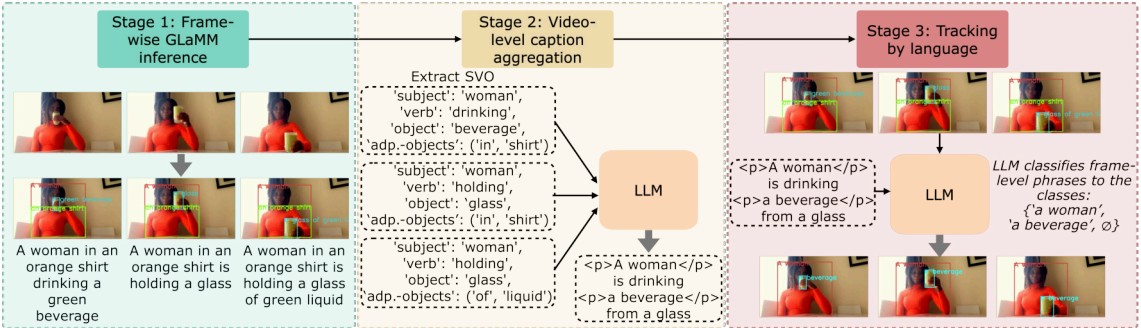

Figure 2: **A method for automatic annotation of spatio-temporally grounded captions.** In the first stage (left), we apply a still-image grounded caption generation model on individual video frames producing temporally inconsistent outputs. In the second stage (middle), the captions from individual frames are aggregated using an LLM into a single video-level caption describing the most salient actions/objects in the video. Third (right), individual frame-level phrases and bounding boxes are associated over time into a temporally consistent labelling of object bounding boxes over the video.

### 4.1 AUTOMATIC ANNOTATION METHOD

We describe our method for generating a pseudolabelled dataset for grounded video caption generation. Given an unlabelled dataset of videos depicting humans interacting with objects or other humans, the goal of the method is to generate both video-level captions describing what is happening in the video *and* bounding boxes grounded to the phrases from the caption that describe the main objects in the video. We leverage foundation

LMMs and LLMs as they have been pretrained on large-scale datasets and are rich sources of information. Our method consists of three steps, as shown in Figure 2: i) frame-wise grounded caption generation, ii) video-level caption aggregation and iii) tracking by language. We describe these steps next.

**Stage 1: Frame-wise grounded caption generation**. We start by applying grounded caption generation, which is the task of generating a text description for the input data along with predicting bounding boxes for the parts of the sentence (phrases) that describe the objects present in the images/videos. Since there are no available video models for this task, we can utilise image-based models and run them in a frame-by-frame basis. We adopt GLaMM (Rasheed et al., 2024), as it has shown very good performance in image-based grounded video captioning. GLaMM generates grounded segmentation masks which we convert to bounding boxes. An example of the outputs of this step can be seen in Figure 2, left. It can be seen that GLaMM generates temporally inconsistent captions across frames, since it does not have access to video-level information. We address this with the next two steps of our proposed method.

**Stage 2: Video-level caption aggregation**. Given the frame-level captions from the previous stage, we wish to obtain a video-level caption describing the most salient actions/objects in the video. We also wish to segment phrases from the caption that correspond to the objects appearing in the video, given the frame-level phrases. These will constitute the labels of interest, which will be consistent across the frames of the video, resolving the language inconsistency of stage 1. We achieve this by prompting an LLM, namely Llama-2 (Touvron et al., 2023), as described next.

GLaMM generates long captions which may contain unnecessary details that can distract the LLM. To address this, we first extract Subject-Verb-Object triplets (SVO) from the frame-level captions as well as adpositions and adpositional objects using Part-of-Speech (POS) tagging. The input to the LLM are the SVO triplets for each frame of the video. By doing so, we feed the LLM with visual knowledge, as seen by the grounded captioning model, that represents the subjects, the actions that they are performing and the objects that are used to perform the action or the objects to which the action is applied. We perform in-context learning with the LLM by feeding it with example pairs of frame-level SVO triplets and the expected video-level captions associated with them. Please see the Supplementary material for the in-context learning prompt that we provide to the LLM. Then, given a new SVO triplet, the LLM answers with the predicted caption for the corresponding video and also tags the phrases that correspond to objects of interest within <p></p> tags. An overview of this process is depicted in Figure 2, middle.

**Stage 3: Tracking by language**. While Stage 2 provides a consistent video-level caption, the phrases that correspond to objects are inconsistent across frames as we have used an image-based model to obtain them. To address this issue, we propose to associate the frame-level phrases with the video-level phrases. We name this procedure *tracking by language*, as we use only the textual description of the phrases without any visual information from the area within the corresponding bounding boxes, and the result is consistent labelling of the bounding boxes throughout the video using the video-level phrases as the labels. Then, bounding boxes with the same label across frames are grouped into video object tracks. We formulate this as a classification problem and prompt again the LLM to solve it with in-context learning. The in-content examples fed to the LLM consist of the input frame-level phrase to be classified and the video-level phrases that make the set of classes for the given video. Please see the Supplementary material for the prompt used for this step. A summary of this process is visualised in Fig 2, right.

With the completion of all three stages, we obtain videos automatically annotated with captions and grounded bounding boxes along with temporally coherent labels that correspond to the phrases from the caption. Additional details and clarifications of our automatic annotation method can be found in Sec. **??** in the Appendix.

## 4.2 HOWTOGROUND DATASET FOR GROUNDED VIDEO CAPTION GENERATION

We introduce HowToGround, an automatically annotated dataset obtained by applying our automatic annotation method to Internet instructional videos from the HowTo100M (Miech et al., 2019) dataset. The HowToGround dataset consists of videos and automatically annotated captions along with temporally consistent bounding boxes across frames grounded to phrases from the captions.

**Data Generation**. We use HowTo100M (Miech et al., 2019) as the video source, a dataset of 100M narrated instructional videos from YouTube, where the humans in the video narrate the actions that they are performing. We choose HowTo100M due to its diversity of actions, scenes, objects and lighting conditions. As in most cases the videos have been captured non-professionally by users, the videos have large viewpoint changes and camera movements, as well as abrupt shot changes. The videos are also of low spatial resolution. These factors along with the diversity of actions, which leads to a long-tailed distribution of events, make the data challenging for grounding.

The narrations of users in HowTo100M have been transcribed by Miech et al. (2019) using ASR, providing text and narration timestamps along with the videos. One might naturally question the need for Stage 1 (video captioning) of our automated curation method, since there is available narrated text. However, the available text is noisy and not suitable for grounding, as in many cases the narrator may thank the viewers, commercials might be part of the narration as well as other instances where the narration is not about the visual environment and its associated objects, *e.g.* instructions which cannot be grounded in the video. Moreover, the timestamps are noisy and do not always correspond to the performed action (see Han et al. (2022)). To alleviate this, we use the timestamps from HowToCaption (Shvetsova et al., 2023), where the authors prompted LLMs to aggregate the narrations to generate captions and predict more meaningful timestamps for the associated predicted captions.

We randomly sample 50k video clips from HowTo100M videos using start/end timestamps from HowToCaption. The videos from HowTo100M have variable frame rates usually ranging in 25-30 fps, and we downsample them at 5fps. The majority of the clips are 8 seconds long with a spatial resolution of $455 \times 256$ pixels. We run our automated annotation pipeline on this set of data to obtain the HowToGround dataset.

**Resulting dataset.** After rejecting videos for which our proposed automatic annotation method failed, *e.g.* the LLM has not provided the expected type of answer, we end up with 48.5k automaticaly annotated videos. We use these videos along with the automatic annotations to train the VideoGLaMM grounded video caption generation model, which we describe next.

## 5 VIDEOGLAMM: A MODEL FOR GROUNDED VIDEO CAPTION GENERATION

We propose VideoGLaMM, a video LMM for the GROC task, see Figure 3. We take inspiration from the GLaMM (Rasheed et al., 2024) model due to its state-of-the-art performance for image-based Grounded Caption Generation. As video LMMs require large amounts of data for training, we build on a pretrained version of GLaMM. The *novel components* of our proposed VideoGLaMM model are (shown in dashed red rectangles in Figure 3): i) the **spatio-temporal adapters with pooling** which enable modelling temporal information efficiently, ii) the **bounding box decoder** which allows to re-use the large-scale pretrained decoder weights of GLaMM and iii) the **temporal objectness head** for modelling objects that temporally leave the frame. We ablate these components in Table 2 and demonstrate their key importance.

**Overview of the architecture.** Figure 3 shows the different components of our approach. The Global Video Encoder, $\mathcal{V}_e(\cdot)$, outputs video features, $o_e$, which are pooled spatio-temporally, resulting in the video prompts. These are projected to a language embedding space with $VL(\cdot)$. A prompt consisting of video-language tokens is ingested by the LLM, $\mathcal{LM}(\cdot)$, which is prompted to generate a caption for the video by tagging the phrases that correspond to objects and appending them with detection tokens (shown with red and green in

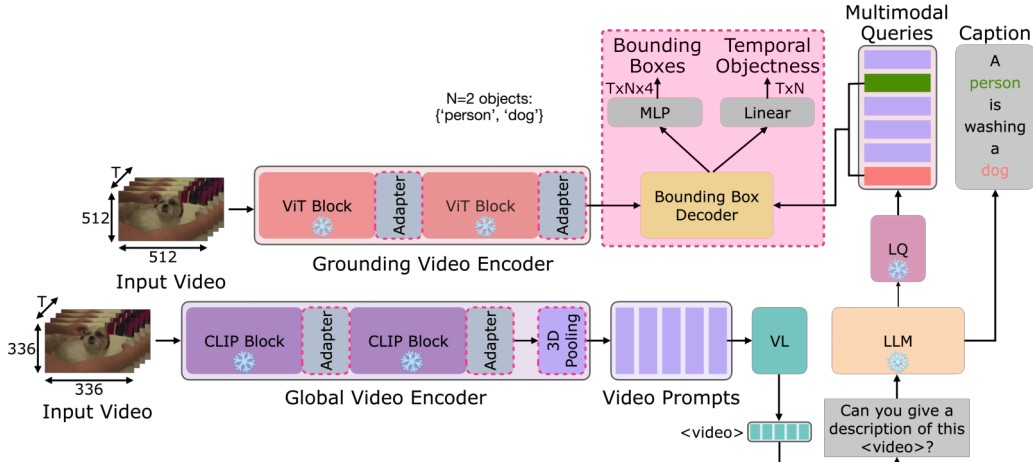

Figure 3: **An overview of our VideoGLaMM grounded caption generation model.** The key technical innovations enabling grounded caption generation in video are outlined by dashed red rectangles and include: (i) spatio-temporal adapters; (ii) the bounding box decoder and (iii) the temporal objectness head.

the LLM's generated caption in Figure 3). The LLM's output hidden states that correspond to the generated caption are projected to queries (using $LQ(\cdot)$). The queries that correspond to detection tokens are fed to the bounding box decoder, $D(\cdot)$. The Grounding Video Encoder, $\mathcal{V}_g(\cdot)$, outputs fine-grained video features, which are also fed to the decoder. The decoder performs cross-attention frame-wise between the queries and the outputs of $\mathcal{V}_g(\cdot)$, $o_g$, which are used as keys/values. Finally, the prediction heads output bounding box predictions and temporal objectness scores for each object at each frame. This objectness score is used to predict the presence/absence of the object in each video frame and is of major importance for the grounded video caption generation task. For details about the visual backbones $\mathcal{V}_e(\cdot)$ and $\mathcal{V}_g(\cdot)$ as well as details about the LLM $\mathcal{LM}(\cdot)$ including the format of its multimodal inputs and its vocabulary, please see Section B in the Appendix.

**Spatio-temporal adapters and pooling**. We obtain $\mathcal{V}_e(\cdot)$ and $\mathcal{V}_g(\cdot)$ by adapting the respective pretrained image encoders of GLaMM. We achieve this by interleaving spatio-temporal adapter layers between the original encoder layers. To stabilise training, we add residual connections and introduce a learnable parameter that is multiplied by the adapter's output and starts from 0 at the beginning of training. By doing so, at the beginning of training the adapter's output is effectively cancelled out and the network observes only the original encoder's output. As training progresses, the learnable parameter is tuned and the network automatically adjusts the contribution of the adapter based on the gradients of the loss. A similar procedure has also been employed by Flamingo (Alayrac et al., 2022) where it has been shown that it helps for a more stable training. An adapter layer performs $a(o) = o + tanh(\alpha) \times f(o)$, where $o$ is the output of the preceding encoder layer, $\alpha$ is the tunable parameter that is initialised to 0 and passes through a $tanh$ activation and $f(\cdot)$ is the adapter layer. As feeding the full video tokens $o_e$ to the LLM is computationally prohibitive, we introduce a spatio-temporal pooling function after the output of $\mathcal{V}_e(\cdot)$, *i.e.*, $o_p = p(o_e)$.

**Projection layers**. We project the outputs of the Global Video Encoder and the output hidden states of the LLM with MLPs, $o_{p'} = VL(o_p)$ and $o_q = LQ(o_l)$, where $VL(\cdot)$ projects the visual features to an embedded language space, while $LQ(\cdot)$ projects the LLM's hidden states to queries. $o_{p'}$ is the LLM's visual input while $o_q$ is input to the bounding box decoder that is described next.

**Bounding box decoder and prediction head**. We re-purpose the decoder of GLaMM for bounding box decoding. This allows us to leverage the pretrained weights of the decoder. GLaMM's decoder follows the

Table 1: Performance comparison of the proposed VideoGlaMM model, the proposed pseudolabelling method and the GLaMM (Rasheed et al., 2024) baseline on our human-annotated test set for the Grounded Video Caption Generation task. In the frame-level set-up performance metrics are calculated across all frames of all videos in a similar manner as done in a still-image evaluation. In the video-level set-up the metrics are calculated per video and averaged across videos. Improvement / decline in comparison to GLaMM shown in parenthesis.

| | Method | METEOR | CIDER | AP50 | mIOU | Recall |
|---|---|---|---|---|---|---|
| Frame-level | GLaMM | 11.9 | 29.9 | 20.8 | 13.2 | 19.6 |
| | Pseudolabelling | 13.8 (▲ 1.9) | 40.0 (▲ 10.1) | 20.6 (▼ 0.2) | 15.1 (▲ 1.9) | 18.2 (▼ 1.4) |
| | VideoGLaMM | **14.2** (▲ 2.3) | **46.8** (▲ 16.9) | **25.2** (▲ 4.4) | **17.5** (▲ 4.3) | **21.9** (▲ 2.3) |
| Video-level | GLaMM | 11.9 | 29.9 | 26.6 | 13.1 | 22.0 |
| | Pseudolabelling | 13.8 (▲ 1.9) | 40.0 (▲ 10.1) | 27.1 (▲ 0.5) | 15.1 (▲ 2.0) | 20.4 (▼ 1.6) |
| | VideoGLaMM | **14.2** (▲ 2.3) | **46.8** (▲ 16.9) | **33.7** (▲ 7.1) | **17.4** (▲ 4.3) | **24.6** (▲ 2.6) |

decoder architecture of SAM (Kirillov et al., 2023) and is designed for segmentation mask decoding. It uses the visual features of the Grounding Image Encoder as queries and the embedded segmentation tokens as keys/values and applies cross-attention to them, resulting in an output of the same dimensionality as the input visual features that is used for predicting the masks. We transform the mask decoder to a bounding box decoder by using the embedded detection tokens as queries, and the visual features of the Grounding Video Encoder as keys/values, resulting in an output that has same length as the detection tokens, allowing us to predict a bounding box for each detection token that corresponds to a noun phrase in the caption. Importantly, while $\mathcal{V}_g(\cdot)$ performs video processing, we apply the cross-attention in a frame-wise fashion to predict objects at each frame of the input video. Formally, the bounding box decoder performs: $o_d = D(o_g, \mathbb{1}_{\{o_q=<DET>\}})$, where $o_d \in \mathbb{R}^{T \times N_d \times D}$, $N_d$ is the number of detection tokens predicted by the LLM, $D(\cdot)$ is the decoder, and $\mathbb{1}_{\{o_q=<DET>\}}$ is an indicator function selecting only the embedded language tokens, $o_q$, that correspond to detection tokens. We employ a bounding box prediction head on the output of the decoder, $o_d$. It is an MLP that predicts bounding box coordinates for the embedded detection tokens at each frame: $p_{bb} = h_{bb}(o_d)$, where $p_{bb} \in \mathbb{R}^{T \times N_d \times 4}$ are the bounding box predictions and $h_{bb}(\cdot)$ is the bounding box head.

**Temporal objectness head**. As discussed before, one major challenge for videos is that objects might disappear and reappear in different frames of the video. To address this, we introduce a *temporal objectness head*. Different than objectness predictions in image-based object detection, the purpose of this head is to predict whether an object is visible or not at a given frame of a video: $p_{tobj} = h_{tobj}(o_d)$, where $p_{tobj} \in \mathbb{R}^{T \times N_d \times 1}$ are the temporal objectness scores and $h_{tobj}(\cdot)$ is the temporal objectness head. During inference, we threshold $p_{tobj}$ and for each frame we select only the bounding boxes for which the temporal objectness scores pass the threshold.

**Loss function**. Our loss function is a combination of a language modelling loss for captioning, two spatial losses relevant to video object detection and a temporal objectness loss. Their details can be found in Section B.

## 6 EXPERIMENTS

**Implementation details**. All implementation details including architectural choices of VideoGLaMM as well as training/inference details can be found in Section B in the Appendix.

**Results**. The results on our human annotated test set are presented in Table 1. We compare results of the proposed VideoGLaMM model with the pseudolabelling method (section 4) and (still image-based) GLaMM by running the pseudolabelling and GLaMM on the test set. The pseudolabelling method is a natural fit for a

baseline for this task as it performs image-based grounded captioning followed by video-level aggregation with LLMs without any training. Moreover, GLaMM helps to assess the benefits of our method in comparison to still-image grounding. For GLaMM, we select the caption of the center frame of each video as the video-level caption.

This evaluation provides two useful insights. First, our pseudolabelling method improves over GLaMM, where the improvement is significant for captioning with the pseudolabelling method obtaining 10.1 points higher CIDER score. This demonstrates the importance of the video-level caption aggregation (2nd stage of the pseudolabelling). The recall of our pseudolabelling decreases in comparison to GLaMM. This is natural because GLaMM predicts labels for the bounding boxes independently per frame, increasing the chance of correctly matching the ground truth labels in each frame, leading to a higher number of true positives (as discussed in Section 3, recall considers both predicted bounding boxes and embeddings of predicted labels). Nevertheless, this comes at the cost of temporally inconsistent labelling and therefore inability to track objects, which is resolved by the 3rd stage of our pseudolabelling pipeline.

Second, VideoGLaMM advances the results further with significant margins across the board. Importantly, VideoGLaMM outperforms significantly vanilla GLaMM on AP50 (7.1 points in video-level and 4.4 points in frame-level evaluation) and mIoU (4.3 points); while it improves in recall over the pseudolabelling method that suffers in this metric (4.2 points and 3.7 points in video-level and frame-level evaluation respectively). Lastly, VideoGLaMM performs better in captioning too. These results indicate that there is a subtle interplay between our proposed pseudolabelling method and VideoGLaMM which is critical for improving the captioning and grounding accuracy of the model as well as the ability of the model to predict the presence of objects in the video both as a natural language description and as a spatio-temporally grounded bounding box. Components of VideoGLaMM that contribute substantially towards this are: i) the temporal objectness, ii) the adapters, iii) freezing the pretrained models to maintain the rich knowledge acquired from GLaMM pretraining while fine-tuning only key parts of the model. We ablate these choices next.

**Ablations**. Tables 2 and 3 compare variants of our VideoGLaMM model. In particular, in Table 2 we ablate the adapters, the temporal objectness and unfreezing parts of the model. The importance of each of those components is evident as removing each component causes performance degradation. Specifically, unfreezing key parts of the architectures and employing the adapters have a significant impact on the model's performance particularly for CIDEr, AP50 and recall. We can conclude that spatio-temporal modelling using the adapters is important for grounded video caption generation, probably because it allows to learn holistic video semantics through the Global Video Encoder and to produce spatio-temporally consistent representations through the Grounding Video Encoder. Last but not least, temporal objectness provides great benefits in AP50 and mIoU, demonstrating the significance of modelling objects that temporally leave the frame in our GROC dataset. This is at the expense of recall – turning off the temporal objectness is equivalent to predicting bounding boxes for each object in the caption in every frame, retrieving more positive instances while introducing a significant number of false positives as it is evident by AP50. In Table 3, we examine which parts of the model are beneficial to be fine-tuned. We opt to keep the visual backbones and LLM frozen to reduce overfitting and retain the rich knowledge learnt from GLaMM pretraining. Note, that in every case we fine-tune the embedding and output layers of the LLM since we modify its vocabulary with special tokens, as well as the bounding box and temporal objectness heads as they are necessary for the grounded video caption generation task. We observe that fine-tuning the decoder and VL projection layer is beneficial while fine-tuning the LQ layer in addition results in slightly worse performance. That is probably because the decoder is of central importance for detection while the VL projection transforms the video representation in a format that is comprehensible by the LLM. Additional ablation regarding the automatic annotation approach to obtain the pseudolabels is in the appendix (Table 7, Section D) and demonstrates that our automatic annotation method is general enough and robust under different grounding models to generate frame-level label (Stage 1).

**Qualitative Results**. We show qualitative results of VideoGLaMM on two example videos in Figure 4. We showcase three important properties of VideoGLaMM. First, VideoGLaMM grounds multiple objects in both

Table 2: Ablations of model components: Video-level evaluation on the validation set. AD: adapters, TO: temporal objectness.

| Unfreeze | AD | TO | METEOR | CIDEr | AP50 | mIOU | Recall |
|---|---|---|---|---|---|---|---|
| ✗ | ✓ | ✓ | 12.6 | 53.5 | 32.2 | 18.0 | 23.8 |
| ✓ | ✗ | ✓ | 12.8 | 50.8 | 32.3 | 18.6 | 23.7 |
| ✓ | ✓ | ✗ | 13.4 | 57.7 | 30.9 | 15.6 | **27.0** |
| ✓ | ✓ | ✓ | **13.4** | **57.7** | **36.3** | **18.9** | 25.1 |

Table 3: Combinations of unfreezing various parts of VideoGLaMM: Video-level evaluation on the validation set. B Decoder: Box Decoder.

| B Decoder | VL | LQ | METEOR | CIDEr | AP50 | mIOU | Recall |
|---|---|---|---|---|---|---|---|
| ✗ | ✓ | ✓ | 12.8 | 54.8 | 30.4 | 17.7 | 22.5 |
| ✓ | ✗ | ✓ | 12.4 | 50.1 | 35.6 | 18.2 | **25.3** |
| ✓ | ✓ | ✗ | **13.4** | 57.7 | **36.3** | **18.9** | 25.1 |
| ✓ | ✓ | ✓ | 13.2 | **59.3** | 35.9 | 18.8 | 24.8 |

examples. Second, the model produces spatio-temporally smooth predictions even under viewpoint changes, as shown in the example on the top. Finally, in the bottom example we demonstrate that temporal objectness models objects that temporally leave the frame, as it does not predict a bounding box for the hand in the third and fofth frame where the hand disappears.

# 7 CONCLUSION

We have proposed a new task, grounded video caption generation, together with a new manually annotated test set, which we refer to as GROC. We have also introduced a new automatic annotation method and pseudo annotated a large-scale training dataset of instructional videos, which we call HowToGround. We have designed a new grounded video caption generation model and trained the model on the HowToGround dataset. The obtained results validate the new training dataset and model by demonstrating their benefits over still-image grounding baselines setting state of the art for the new task of grounded video caption generation. We believe the new training and test datasets can spark further interest of the research community in this problem.

**Limitations.** The model may inherit biases of the still image GLaMM model employed to generate our automatic pseudo annotations and also used for initialization. Combining the automatic pseudo annotations together with additional manual training annotations would open up the possibility for Internet-scale training of grounded video caption generation models.

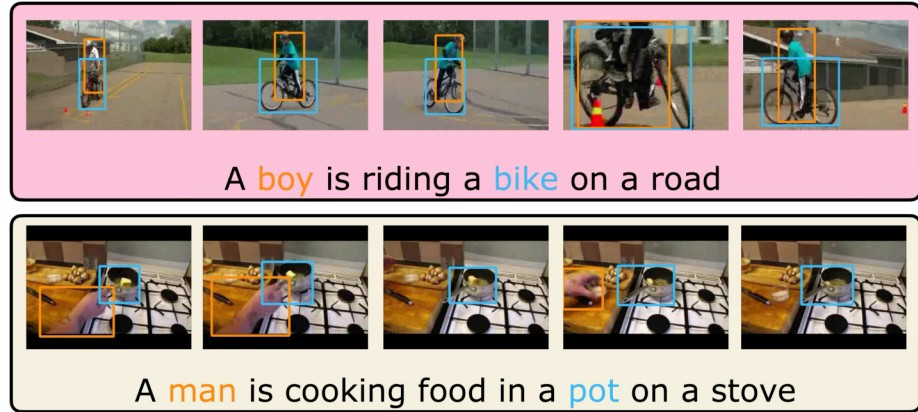

Figure 4: Qualitative examples showing the predictions of VideoGLaMM on two videos. The examples showcase three important properties of VideoGLaMM: i) it can ground multiple objects (both), ii) it produces spatio-temporally consistent predictions (top), iii) temporal objectness models objects that temporally leave the frame (bottom, the hand disappears in the third and fifth frames).

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

APPENDIX

## A DATASET STATISTICS AND COMPARISON WITH OTHER DATASETS

### A.1 DATASET STATISTICS

Table 4 reports the statistics of both the training pseudolabelled dataset and the human annotated test set. Word clouds of the natural language descriptions for the training pseudolabelled dataset and the human annotated test set are shown in Figure 5.

Table 4: Statistics of HowToGround and GROC datasets.

| Statistic | HowToGround | GROC |
|---|---|---|
| Avg num frames | 44.6 | 40.2 |
| Avg duration (seconds) | 7.9 | 8.0 |
| Avg num instances per video | 61.9 | 122.1 |
| Total num instances | 3,023,050 | 12,090 |
| Avg box width $\times$ height | $249.2 \times 179.9$ | $174.4 \times 141.9$ |
| Avg tube length (frames) | 9.1 | 29.3 |
| Avg caption length (words) | 12.8 | 13.1 |

### A.2 COMPARISON WITH OTHER DATASETS

Tables 5 and 6 compare our HowToGround and GROC datasets, respectively, with other datasets across various axes. Since HowToGround is designed for training, in Table 5 we compare it with datasets that have training sets. In the same manner, since we propose GROC as a test set, in Table 6 we compare it with datasets that provide a test set.

Both HowToGround and GROC are the only datasets that simultaneously have multiple annotated objects per frame, annotations across multiple frames and annotated noun phrases from the caption that are linked to object bounding boxes, making them the only suitable datasets for the proposed grounded video caption generation task. In Table 5, the only other dataset with multiple objects per frame and annotated noun phrases is ActivityNet-Entities. Nevertheless, ActivityNet-Entities provides annotations for a single frame per video segment while HowToGround's training set has an average of 43.5 annotated frames per video segment. GROC is the only test dataset with multiple annotated objects per frame and annotated noun phrases, while it provides the longest captions and has the second largest number of total annotated instances.

Table 5: Comparison of the proposed HowToGround dataset with other training datasets for grounding.

| Dataset | Avg. caption length | Avg. annot. frames | Multiple objects | Phrases | Videos | Total annot. instances |
|---|---|---|---|---|---|---|
| VidSTG (Zhang et al., 2020d) | 10.1 | 275.0 | ✗ | ✗ | 36.2K | 9.9M |
| HC-STVG (Tang et al., 2021b) | 17.4 | 147.2 | ✗ | ✗ | 10.1K | 1.5M |
| ActivityNet-Entities (Zhou et al., 2019) | 13.8 | 1.0 | ✓ | ✓ | 37.4K | 93.6K |
| HowToGround (Ours) | 12.1 | 43.5 | ✓ | ✓ | 48.5K | 3.9M |

## B DETAILS OF THE VIDEOGLAMM MODEL

**Backbones**. VideoGLaMM consists of two video encoders and a multimodal LLM as its main backbones. The Global Video Encoder $\mathcal{V}_e(\cdot)$, takes as input a video $v \in \mathbb{R}^{T \times H1 \times W1}$ and produces an output

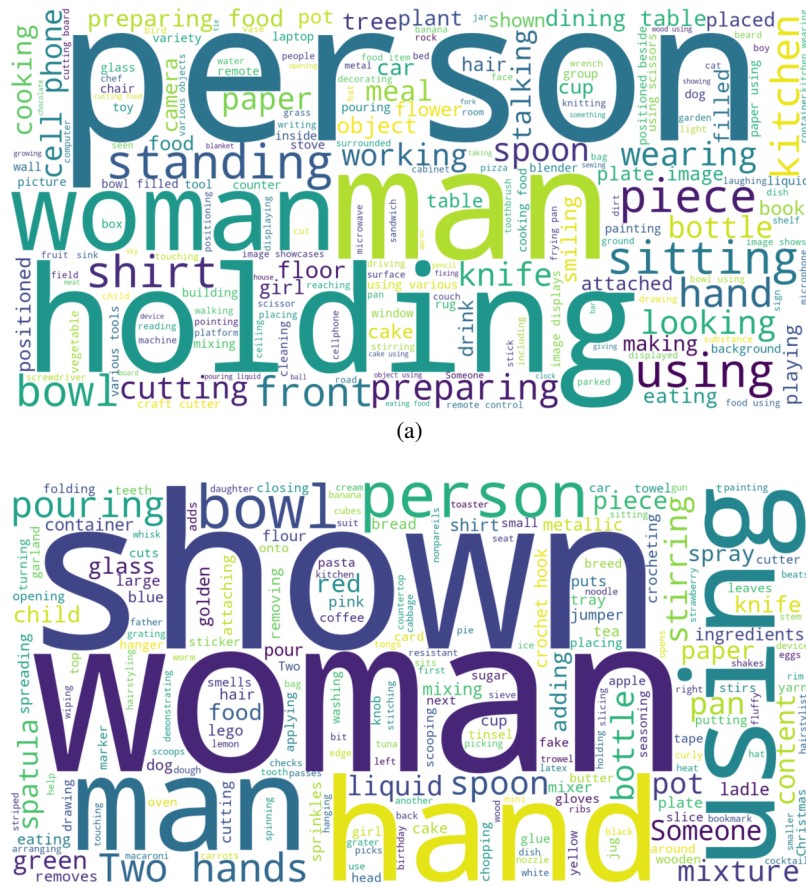

(a)

(b)

Figure 5: Word cloud for (a) pseudolabelled dataset and (b) human-annotated test set.

Table 6: Comparison of the proposed GROC dataset with other test datasets for grounding.

| Dataset | Avg. caption length | Avg. annot. frames | Multiple objects | Phrases | Videos | Total annot. instances |
|---|---|---|---|---|---|---|
| VidSTG (Zhang et al., 2020d) | 10.1 | 262.3 | ✗ | ✗ | 4.6K | 1.2M |
| YouCook-Interactions (Tan et al., 2021) | 8.6 | 9.8 | ✗ | ✗ | 0.25K | 2.5K |
| GroundingYT (Chen et al., 2024) | 2.0 | 4.1 | ✗ | ✗ | 4.2K | 17.3K |
| GROC (Ours) | 13.2 | 43.2 | ✓ | ✓ | 1K | 119K |

$o_e \in \mathbb{R}^{T \times \frac{H1}{p} \times \frac{W1}{p}}$, where $p$ is the patch size of the underlying visual transformer. Its purpose is to provide a holistic representation of the video that will be ingested by the LLM. The Grounding Video Encoder $\mathcal{V}_g(\cdot)$, takes as input a video $v \in \mathbb{R}^{T \times H2 \times W2}$, where $W2 > W1$ and $H2 > H1$. It produces $o_g \in \mathbb{R}^{T \times \frac{H2}{p} \times \frac{W2}{p}}$. $o_g$ is used to ground phrases from the caption to the visual content, which is performed by the bounding box decoder that is described later. The input video to the Grounding Video Encoder is of larger spatial resolution than that of the Global Video Encoder for enhanced localisation capability. Finally, the LLM $\mathcal{LM}(\cdot)$ takes as input a multimodal sequence $s \in \mathbb{R}^{L \times D}$ and produces an output $o_l$ of the same size. Its input is of the form `The <video> provides an overview of the`

```
video.  Could you please give me a description of the video?  Please
respond with interleaved bounding boxes for the corresponding parts of
the answer.
```
<video> is replaced by the output of $\mathcal{V}_e(\cdot)$, and therefore the LLM ingests mixed language and visual tokens. We also augment the LLM's vocabulary with a detection token `<DET>`, prompting the model to generate responses with `<DET>` tokens by the phrases that correspond to objects to be detected in the video.

**Loss function**. Our loss function is a combination of a language modelling loss and losses relevant to video object detection. The language modelling loss is a Cross-Entropy loss applied on $o_l$. For object detection, we follow DETR (Carion et al., 2020) and use a gIoU loss (Rezatofighi et al., 2019) and an L1 loss applied on $p_{bb}$. Different than Carion et al. (2020), the losses are applied per frame and summed over frames. Moreover, the losses are applied only to the objects that appear in the frame (rather than each object in the caption) using the ground-truth temporal objectness scores. The representation that we use for the bounding boxes is `[x,y,w,h]` and their coordinates are normalised with the dimensions of the video. Finally, we employ a binary cross-entropy loss on $p_{tobj}$. Our loss is, hence, defined as:

$$\mathcal{L}_{LM} = CE(o_l), \quad \mathcal{L}_{gIoU} = gIoU(p_{bb}, gt_{bb}), \quad \mathcal{L}_{L1} = L1(p_{bb}, gt_{bb}), \quad \mathcal{L}_{tobj} = BCE(p_{tobj}, gt_{tobj}) \tag{1}$$

$$\mathcal{L} = \lambda_{LM} \times \mathcal{L}_{LM} + \lambda_{gIoU} \times \mathcal{L}_{gIoU} + \lambda_{L1} \times \mathcal{L}_{L1} + \lambda_{tobj} \times \mathcal{L}_{tobj}, \tag{2}$$

where $gt_{bb}$ are the ground truth boxes and $gt_{tobj}$ are the ground truth objectness scores and $\lambda$ are the weights for the losses.

**Training/inference**. We realise the Global Video Encoder $\mathcal{V}_e(\cdot)$ with a CLIP-L (Radford et al., 2021) model with an input of 336×336 and a patch size of 14. The Grounding Video Encoder $\mathcal{V}_g(\cdot)$ is instantiated with a SAM (Kirillov et al., 2023) encoder and the bounding box decoder $D(\cdot)$ is a SAM-based decoder, the same as in GLaMM (Rasheed et al., 2024). The LLM $\mathcal{LM}(\cdot)$ is a Vicuna-7B model (Chiang et al., 2023). During training we keep $\mathcal{V}_e(\cdot)$, $\mathcal{V}_g(\cdot)$ and $\mathcal{LM}(\cdot)$ frozen. $\mathcal{V}_g(\cdot)$ originally takes as input 1024× 1024 images. As this is too large to fit in memory for videos, we instead use 512×512 video spatial resolution, while we interpolate the positional encodings of $\mathcal{V}_g(\cdot)$ and fine-tune them. Adapters are 3D spatiotemporal convolutional layers with a kernel of size $3 \times 3 \times 3$ and a stride of 1. We apply adapters to every 3 layers of $\mathcal{V}_e(\cdot)$ and to all global attention layers of $\mathcal{V}_g(\cdot)$. The bounding box head $h_{bb}$ is an MLP with two FC layers and a ReLU activation function in between, while the temporal objectness head $h_{tobj}$ is a linear layer. Both prediction heads employ a sigmoid activation function. We apply a threshold of 0.5 to the temporal objectness scores. Both the adapters and the prediction heads are randomly initialised. We use $T = 8$ frames for the videos during both training and testing. During training we perform random sparse sampling of frames by splitting the video in 8 segments and randomly drawing a frame from each segment while during testing we pick the centre frame of each segment.

We train VideoGLaMM for 10 epochs using a batch size of 128. We use a learning rate of $10^{-4}$ with warmup for the first 100 training steps and linearly decay the learning rate for the rest of training. We do not apply any weight decay or spatial data augmentation. We use $\lambda_{LM} = \lambda_{gIoU} = \lambda_{L1} = \lambda_{tobj} = 1$.

## C   DETAILS OF THE AUTOMATIC ANNOTATION METHOD

**Multiple people in the video**. Our automatic annotation method can handle multiple subjects in a video as long as one of the two following conditions are met: a) the subjects are described with a distinct language, *e.g.* 'man with green jumper' and 'man with blue shirt', or b) the subjects are within a Subject-Verb-Object relationship even when described with the same terms, *e.g.* ('person', 'dances', 'with', 'person') which would produce 'A person dances with another person'. If neither conditions are met, the caption aggregation (Stage 2) may merge the two subjects into one.

Table 7: Comparison of our pseudolabeling approach with the proposed Stage 1 for grounding vs. an alternative approach for grounding in Stage 1 based on GIT (Wang et al., 2022), Llama3 (Dubey et al., 2024) and OWLv2 (Minderer et al., 2024).

| Method | METEOR | CIDER | AP50 | mIOU | Recall |
|---|---|---|---|---|---|
| Pseudolabelling w. proposed Stage 1 | 12.5 | 43.8 | **23.4** | 15.5 | **18.8** |
| Pseudolabelling w. alternative Stage 1 | **12.9** | **45.0** | 18.9 | **16.9** | 14.3 |

**Association of verbs and objects** is naturally performed through the Subject-Verb-Object triplets. For example, given two relationships: ('man', 'cuts', 'onions') and ('woman', 'stirs', 'food', 'in', 'pot'). The LLM-based caption aggregation step (Stage 2) has sufficient information to associate the man with the action of cutting the onions and the woman with stirring the food.

**Additional details of Stage 3**. We provide additional details of the procedure of Stage 3 using the example from Fig. 2, right. The object in the woman's hands is described as 'a green beverage', 'a glass', and 'a glass of green liquid' across different frames. Stage 2 has provided the video-level noun phrases 'a woman' and 'a beverage'. Stage 3 is formulated as a classification problem where each one of 'a green beverage', 'a glass', and 'a glass of green liquid' are the inputs to be classified in one of the classes {'a woman', 'a beverage', $\emptyset$} and thus associated with the right bounding box. The class $\emptyset$ represents the "None" class, *i.e.* when an input does not belong to any of the known classes and it is useful for noisy inputs.

# D  ADDITIONAL EXPERIMENTS

We show below that our pseudolabeling approach is general enough and works with different grounding models to generate frame-level labels. This is important as it can increase the variability of the outputs and can make the output less prone to biases from a single frame-level grounding model.

To demonstrate the generality of our pseudolabelling method, we have replaced GLaMM from Stage 1 with an alternative grounding method. To this end, we've incorporated a GIT approach (Wang et al., 2022) for captioning, Llama3 (Dubey et al., 2024) for extraction of noun phrases from the caption, and the open vocabulary detector OWLv2 (Minderer et al., 2024) for detecting and localizing the noun phrases in a given frame. Similarly to the GLaMM approach, the outputs of Stage 1 are: a) frame-level captions, b) groundable noun phrases from the captions and c) bounding boxes for each phrase. The quantitative results of this new frame-level grounding approach can be found in Table 7 where we compare the performance of the original pseudolabelling method with this alternative on the validation set. We observe that overall this alternative pseudolabelling method maintains a reasonable performance, demonstrating that the subsequent steps of our approach can handle outputs of different pseudolabelling methods fairly well. Nevertheless, pseudolabelling with our proposed Stage 1 performs noticeably better in AP50 and Recall, as GLaMM has been explicitly trained for grounding, which is not the case for GIT, Llama3 and OWLv2, which can underperform for various reasons, including Llama3 extracting non-groundable noun phrases or OWLv2 missing objects. At the same time, pseudolabelling with the alternative Stage 1 performs slightly better for the captioning metrics and mIoU due to the superior performance of GIT for captioning and OWLv2 for object detection.

Given the complementary benefits of the proposed and alternative Stage 1, one can improve the pseudolabels by combining the outputs of the two approaches, which we leave for future work.

## E    QUALITATIVE RESULTS

Figure 6 shows examples of pseudolabelling annotations on the training set using our proposed automatic annotation method (Section 3 in the main paper). Our automatic pseudo-annotation method produces video-level natural language captions describing the main action in the video together with temporally consistent bounding boxes grounding the main objects in the video.

Figure 7 shows qualitative results of our VideoGLaMM model (section 5 in the main paper) on the test set.

## F    PROTOCOL FOR HUMAN ANNOTATIONS

Below we describe the annotation guidelines for annotating our validation/test sets.

---

**Annotation Guidelines:**

1. Video Selection:
   - You will be provided with a larger set of videos than needed.
   - Your first task is to select clips that are considered 'interesting' based on criteria that will be discussed further. An 'interesting' video typically includes dynamic events or actions that are clear and distinguishable despite the low video quality. In those events/actions people usually interact with objects, e.g. 'A man is cutting an onion using a knife'. 'Non-interesting' events are typically static, e.g. a person simply standing/sitting and talking. Non-interesting events are also events with ambiguous actions taking place, i.e. generic/abstract actions that cannot be described concisely or actions for which the annotator is unsure about what is happening in the video.

2. Video Annotation:
   - For each selected video clip, write a concise, one-sentence description of the main event taking place in the clip. If the action is too complex, use at most two sentences for describing it, but prioritise one-sentence descriptions.
   - Focus only on the objects that humans interact with rather than describing densely every object in the scene.
   - To enrich the language descriptions, also describe properties of objects such as color, shape, etc, e.g. 'blue cup' or 'red onion'. It is not strictly necessary to always describe the object's property but only when deemed important by the annotator.
   - When you are unsure about the object being used, you can simply describe it as 'object'. If object is unknown but the category of the object is known, please describe the object using its category, e.g. 'food'.
   - When there are two or more humans in the scene, use one of their characteristics to distinguish them, e.g. 'the woman in the red shirt standing next to the woman in the green shirt is putting a strawberry on a cocktail glass'.
   - If there are multiple actions happening consecutively, describe all of them and their associated objects. E.g. 'a person is doing action-1 using object-1, then doing action-2 with an object-2'. As shown in the example, you can use 'then' for connecting temporally adjacent actions.
   - Provide bounding boxes for humans/different objects mentioned in your description. These bounding boxes should be applied to all frames where the objects are visible.

---

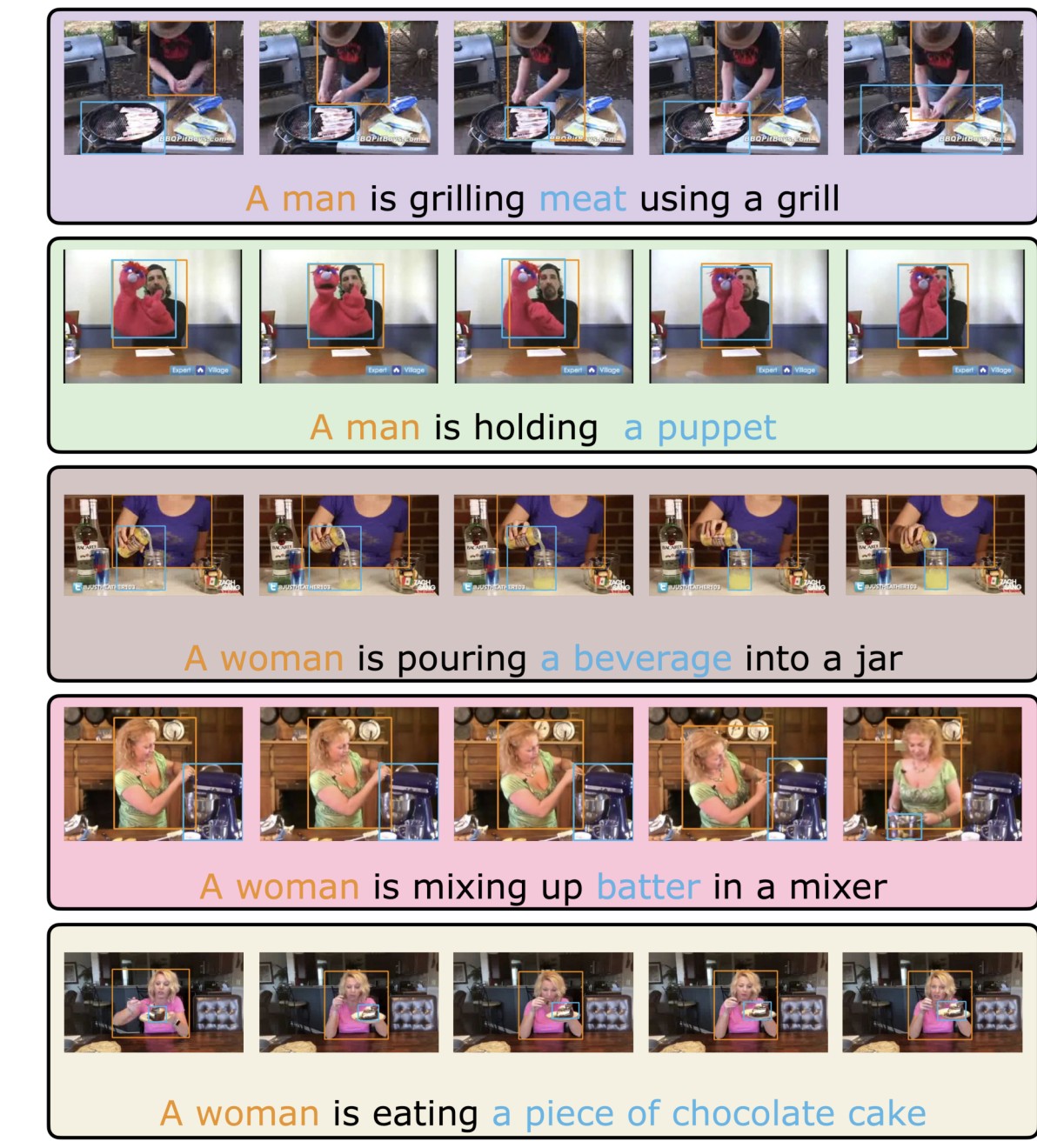

Figure 6: Examples of pseudolabelling annotations on the training set using our proposed automatic annotation method (Section 3 in the main paper). The color coded sentence fragments are spatio-temporally localized in the video with the bounding boxes color coded with the same color. Please note how our automatic pseudo-annotation method produces video-level natural language captions describing the main action in the video together with temporally consistent bounding boxes grounding the main objects in the video.

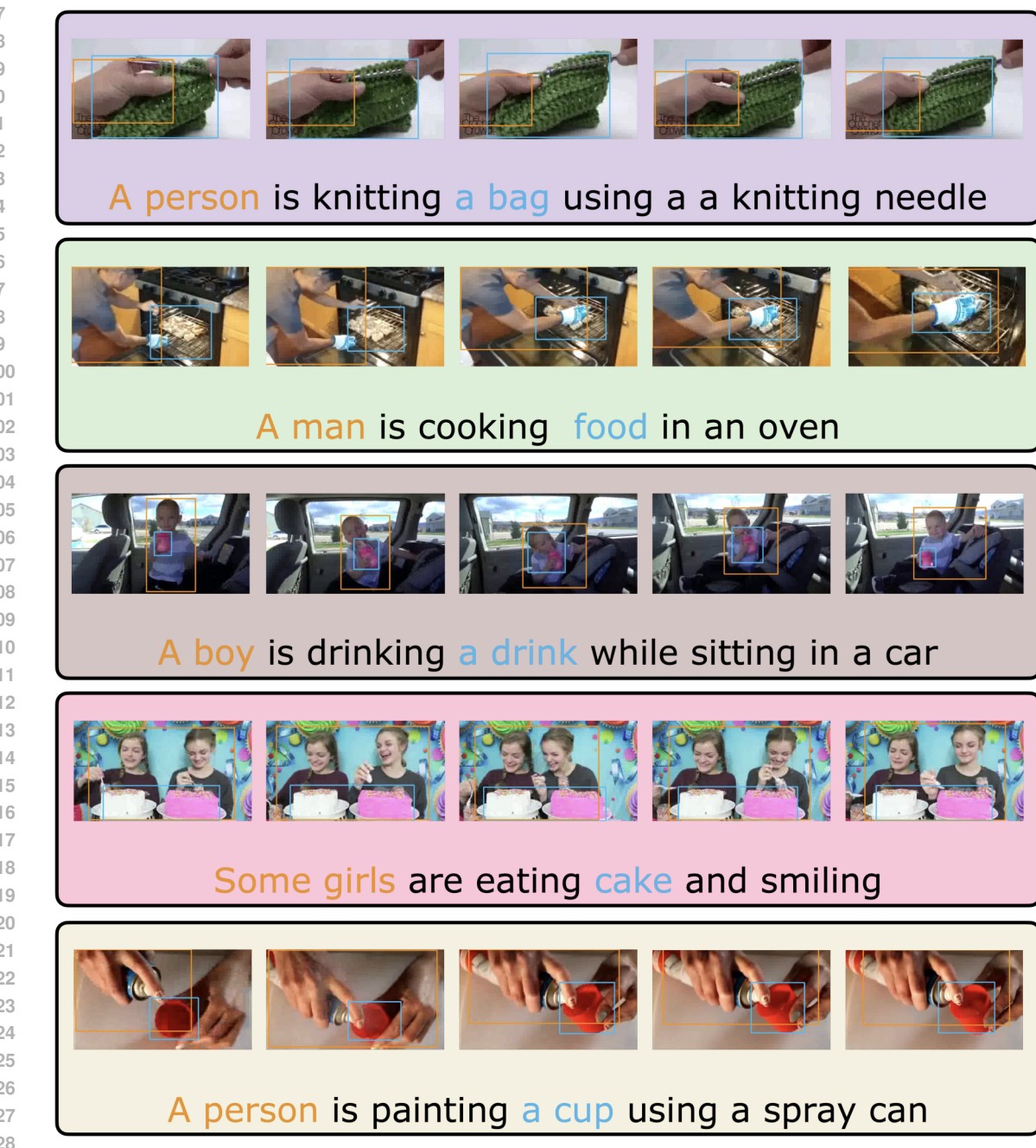

Figure 7: Qualitative results of our VideoGLaMM model (section 5 in the main paper) on the (unseen) test set. The color coded sentence fragments are spatio-temporally localized in the video with the bounding boxes color coded with the same color. Our model is able to produce video-level natural language captions describing the main action in the video together with temporally consistent bounding boxes grounding the main objects in the video.

- Label each bounding box with a short phrase directly from your sentence description (e.g., 'a brown dog', 'persons hands').
- It is not necessary that each object appears in each frame of the video. For example, a person might be using a tool, then leaving it down and using another tool. In this case, you would annotate with bounding boxes the first tool for the first half of the video and the second tool for the second half. Another common case is that objects or the person might disappear and then reappear. In this case, again all instances of the object must be annotated, so you should be careful about objects leaving the scene as they might enter the scene again later.
- If there are many small objects, e.g. mushrooms in a pan, use a single bounding box labelled as 'mushrooms'.
- There are cases where two or more bounding boxes are needed for objects of the same type: a) one bounding box for each human hand when both are used to perform an action, b) one bounding box for each tool/container/appliance etc of the same type that the human is using, e.g. when they are placing food in two dishes, or pouring the content of a shaker in two cocktail glasses.
- Descriptions: Must be accurate and written in fluent English. Suitable for either native speakers or highly proficient English speakers.
- Bounding Boxes: Ensure that bounding boxes accurately encompass the objects for the entirety of their visibility within the clip. The bounding boxes should be consistent and smooth across frames, maintaining size and position as closely as possible given the movement of the object and video quality. An exception is when there are abrupt viewpoint changes of the camera, which might result in objects abruptly changing position and size across neighbouring frames.

The annotation criteria have been extensively discussed with the annotation provider and the annotators have been trained based on those criteria prior to commencing the annotation process. We have also performed a pilot annotation project with the annotation provider on 10 video clips with several rounds of careful checking and feedback. Moreover, the annotation provider performed regular quality reviews on the annotations to ensure that the annotation criteria have been met.

## G    PROMPTS FOR AUTOMATIC CURATION OF SPATIO-TEMPORALLY GROUNDED CAPTIONS

The full prompt for the **Stage 2 (Video-level caption aggregation)** of our pseudolabeling approach (Section 3 in the main paper) is shown in Figure 8 and the full prompt for **Stage 3 (Tracking by language)** in Figure 9.

**System Instructions**

Generate a dynamic, video-level description based on frame-level inputs. The inputs include actions performed in individual frames in the form of Subject-Verb-Object (SVO) triplets along with prepositions and prepositional objects. The SVO triplets describe how actions are performed and how objects interact. Your output should be a concise narrative in 1 sentence, focusing on the most salient actions depicted across the frames. Enclose the exact text of relevant objects within <p></p> tags.

**Input format**:

```
[['subject': 'subject_text', 'verb': 'action_text', 'object': 'object_text',
'prepositions_objects': [('preposition', 'prepositional_object')],],]
```

**Output format**:

```
A Python dictionary with a key 'CAPTION', and as a value a dynamic description of the video content.
```

Infer motion from static descriptions. E.g. 'image shows a person holding a spoon and a bowl' implies 'person is stirring food in a bowl'. Enclose the human and the most frequent object that is used to perform the action within <p></p> tags. If there is no human, enclose the two most frequent objects within <p></p> tags.

**User Input 1**

**SVO**:

```
[['image', 'shows', 'cup'], ['bowl', 'is']],
[['person', 'holding', 'spoon'], ['spoon', 'is', 'bowl'],
['image', 'shows', 'spoon', ('inside', 'bowl')]],
[['person', 'seen'], ['person', 'holding', 'spoon'], ['spoon', 'used'],
['spoon', 'stir', 'food', ('in', 'bowl')]],
[['person', 'holding', 'spoon'], ['spoon', 'is', 'bowl']],
[['person', 'holding', 'spoon'], ['spoon', 'is', 'bowl']],
[['person', 'holding', 'spoon'], ['spoon', 'is', 'bowl']],
['image', 'shows', 'spoon', ('in', 'bowl')]],
[['image', 'shows', 'bottle'], ['bottle', 'positioned', ('beside', 'bowl')]],
[['image', 'shows', 'bottle'], ['bottle', 'positioned', ('beside', 'cup')]],
[['image', 'shows', 'bottle'], ['image', 'placed', ('on', 'counter')],
['bottle', 'positioned', ('beside', 'bowl')]]]
```

**Assistant Response 1**

```
{'CAPTION': '<p>A person</p> is stirring <p>food in a bowl</p> using a spoon'}
```

**User Input 2**

**SVO**:

```
[['hand', 'using', 'cutting board']],
[['woman', 'using', 'cutting board'], ['woman', 'make', 'craft project']],
[['child', 'using', 'craft cutter'], ['child', 'cut', 'object']],
[['child', 'using', 'craft cutter'], ['child', 'cut', 'paper']],
[['woman', 'using', 'craft cutter'], ['woman', 'cut', 'object']],
[['woman', 'using', 'scissors pair'], ['woman', 'cut', 'piece', ('of', 'paper')]],
[['hand', 'using', 'scissors pair'], ['hand', 'cut', 'piece', ('of', 'paper')]],
[['woman', 'using', 'scissors pair'], ['woman', 'cut', 'piece', ('of', 'paper')]],
[['woman', 'using', 'craft cutter'], ['woman', 'cut', 'object']],
[['woman', 'using', 'craft cutter'], ['woman', 'cut', 'plate']]]
```

**Assistant Response 2**

```
{'CAPTION': '<p>A woman</p> is cutting <p>an object</p> using a craft cutter'}
```

**New User Input**

**SVO**: {input_svo}

Figure 8: The full prompt for Stage 2 (Video-level caption aggregation) of our pseudolabeling approach (Section 3 in the main paper).

**System Instructions**

You are tasked with classifying humans and objects to a set of given categories.
**Input format**:

`Human/Object (string), set of categories (lists of strings).`

**Output format**:

`A Python dictionary with a key 'CATEGORY', and as a value the predicted category of the human/object.`

Use 'None' if the human/object doesn't belong to any of the categories. DO NEVER classify a human as the object category and vice versa.

**User Input 1**

**Input**: 'person'
**Categories**: ['a woman', 'her hair']

**Assistant Response 1**

`{'CATEGORY': 'a woman'}`

**User Input 2**

**Input**: 'table'
**Categories**: ['a person', 'a bowl']

**Assistant Response 2**

`{'CATEGORY': 'None'}`

**User Input 3**

**Input**: 'a piece of food on a plate'
**Categories**: ['a woman', 'a meal']

**Assistant Response 3**

`{'CATEGORY': 'a meal'}`

**User Input 4**

**Input**: 'a hand'
**Categories**: ['a person', 'food on a plate']

**Assistant Response 4**

`{'CATEGORY': 'a person'}`

**User Input 5**

**Input**: 'a man in a white shirt and black apron is also present'
**Categories**: ['a person', 'food']

**Assistant Response 5**

`{'CATEGORY': 'a person'}`

**New User Input**

**Input**: {input_object}
**Categories**: {input_categories}

Figure 9: The full prompt for Stage 3 (Tracking by language) of our pseudolabeling approach (Section 3 in the main paper).

