# OpenReview forum: "Grounded Video Caption Generation"
_ICLR.cc/2025/Conference — ICLR 2025 Conference Withdrawn Submission_

### Official Review · Reviewer_KKAm · 2024-10-30

**Soundness:** 2
**Presentation:** 2
**Contribution:** 3
**Rating:** 3
**Confidence:** 3

**Summary:**

This paper proposes a novel task—grounded video caption generation—along with a newly annotated dataset, GROC. The authors introduce a language-based tracking method to create pseudo-annotations for a large-scale instructional video dataset, named, HowToGround, which is then used to train a newly designed model. Finally, they compare the model's performance against still-image grounding baselines, reporting state-of-the-art results in grounded video caption generation.

**Strengths:**

1. Propose a new task: grounded video caption generation, which is significant as large-scale spatio-temporal grounding of natural language is a key step in advancing fields such as human-robot interaction and embodied perception.
2. Propose a manually verified test dataset, which is likely to further drive progress in this field.
3. The model architecture is both innovative and intuitive, with subsequent ablation studies demonstrating the effectiveness of these design choices.

**Weaknesses:**

1. There are concerns regarding the technical soundness of the AUTOMATIC ANNOTATION METHOD section. I understand that the grounded video caption generation task requires high precision; however, due to the richness and ambiguity of language, I am skeptical about the consistency of the language-based tracking method in object localization. For example, while the case of the woman drinking a beverage in the authors' paper is unambiguous, many instances will likely present significant ambiguity. If, in the third frame of Figure 2, the image caption still states 'holding a glass,' how can one determine whether 'beverage' and 'glass' refer to the same object?

2. The writing quality is poor, containing a lot of redundancy, and some phrases read awkwardly and lack fluency. For instance, the phrase 'Differently than the image-based task, where bounding boxes...' could be improved. Additionally, some sentence structures need adjustment, and paragraph transitions require attention, particularly when describing the GROC construction process. There are also typos to address, such as in Section ?? on page 5. Overall, the writing and illustrations need further polishing.

**Questions:**

1. Did the automatic annotation process take into account the ambiguities mentioned in the weaknesses section? Are there any methods in place to reject annotations for data that the model finds ambiguous?

2. Why was SAM2 not considered in the model's structural design? As far as I know, this model targets video and seems more suitable for your task.

3. I hope to see a comparative analysis between the current vLLM and video grounding models in your task, including their differences in performance, efficiency, and application scenarios. This would help in better understanding the strengths and weaknesses of each model and provide guidance for future research directions.

---

### Official Review · Reviewer_c5qs · 2024-11-01

**Soundness:** 2
**Presentation:** 2
**Contribution:** 2
**Rating:** 3
**Confidence:** 3

**Summary:**

The paper introduces a task called grounded video caption generation. It presents a manually annotated benchmark comprising 1,100 video clips, as well as a 50k training set created using an automatic annotation pipeline. GLaMM is adapted with video encoders to achieve GROC. Similar to GLaMM, the authors evaluate VideoGLaMM using a collection of video captioning and object detection metrics. The evaluation results indicate that the proposed method outperforms comparable approaches across various downstream tasks.

**Strengths:**

The paper proposes an interesting and reasonable task of generating captions for videos while simultaneously grounding the corresponding objects.

The paper presents a technically sound framework to achieve GROC.

**Weaknesses:**

I am primarily concerned with the automatic data generation pipeline. In this process, the authors used GLaMM to generate frame-wise captions and then converted these into Subject-Verb-Object (SVO) triplets. Since there is not alignment between generated SVOs, if there are two different instances with the same label in different frames, it is possible that the LLM might treat them as a single instance, potentially leading to inaccurate conclusions and relationships between them.

Similarly, in the tracking by language process, instances are tracking with only textual description of the phrases. This cannot guarantee 2 different instances with same label are mistakenly treated as the single instances. For example, imagining a video that there are 2 different people appearing in different frames but all drinking beers, the process is likely to link them as one person by just considering its textual description. The strategy described in Supp. is not very convincing as there can be 2 different people both with white shirts and drinking.

The process appears to capture only a portion of the video content, potentially overlooking valuable concurrent events within the video clip.

I am also curious about the visual encoders used in VideoGLaMM. Is there any specific reason or necessity for using two different image backbones separately for captioning and bounding box generation?

the reference in Line 230 is missing.

**Questions:**

See above.

---

### Official Review · Reviewer_LZxE · 2024-11-02

**Soundness:** 2
**Presentation:** 1
**Contribution:** 2
**Rating:** 3
**Confidence:** 4

**Summary:**

This paper presents a new task: grounded video caption generation, which aims to generate captions for videos while also providing bounding boxes for the objects mentioned in the captions.
To achieve this, the authors designed an automatic grounded video annotation pipeline. Based on the dataset constructed using this pipeline, the paper trains a model called VideoGLaMM, which performs well on the grounded video caption generation task.

**Strengths:**

- An automated grounded video annotation pipeline was designed, significantly reducing annotation costs.

**Weaknesses:**

- The authors claim to introduced a new task: grounded video caption generation. However, there is prior research on this task, such as PG-VIDEO-LLAVA[1]. How do the authors justify their assertion of novelty?

[1] PG-Video-LLaVA: Pixel Grounding Large Video-Language Models.

- For videos that contain two or more events, how should the automated annotation pipeline be applied? For example, consider a scenario where a man picks up a knife to chop vegetables and then puts the vegetables into a pot. If the knife and pot appear simultaneously in the frame but the action hasn't progressed to the pot yet, how can we avoid generating a bounding box for the pot prematurely?

- The case studies (e.g., Figure 4) presented by the authors seem overly simplistic. Can VideoGLaMM be effectively applied to videos that involve multiple events?

- There are very few comparative methods included in the study. The authors should compare more methods, such as the integration of video captioning models with object detection models, and PG-Video-LLaVA.

**Questions:**

See Weaknesses.

---

### Official Review · Reviewer_Gwsz · 2024-11-02

**Soundness:** 2
**Presentation:** 4
**Contribution:** 2
**Rating:** 5
**Confidence:** 4

**Summary:**

This paper addresses the task of grounded video captioning, where, given a video, a model must generate a caption and provide bounding boxes for the objects appearing in the caption. The authors introduce two datasets for this task: a small, manually annotated test set, and a larger, automatically annotated training set. Finally, they train a grounded video captioning model on this dataset and evaluate it against two baselines.

**Strengths:**

* The paper tackles the important task of grounded video captioning, which has numerous applications in downstream tasks such as video retrieval and open-world object tracking in videos.
* The writing is clear.
* The paper introduces the HowToGround dataset for grounded video captioning and provides a high-quality, manually annotated test set for the task.
* The paper proposes an architecture for grounded video captioning, building upon a pre-trained grounded image captioning model.

**Weaknesses:**

* The first paragraph of the introduction feels more like a related work section, making it hard to follow. I would suggest starting by clearly stating the task you aim to solve and providing a high-level summary of the existing work.
* The method for automatically creating the dataset uses SVOs, which can lead to important details being omitted from captions, such as the color of the shirt in Figure 2. This omission could hinder downstream tasks. For instance, a model aiming to find videos of orange shirts might miss this information, which is otherwise captured in current captioning models like GlaMM.
* The paper’s main contribution is the creation of the HowToGround dataset. However, the choices made during dataset creation are not justified, nor does the paper include any ablation study of the creation steps. Including such a study, showing alternative choices for each step, would greatly strengthen the paper. For example, how about using SAM-2 or an open-world object detector to identify object locations? How would VideoGLaMM perform if step 3 were omitted from the dataset creation process? How well does the resulting VideoGLaMM for different train set sizes?

Typos/Errors:
* L230 - Section reference is broken.

**Questions:**

* L140 mentions "video selection where ‘interesting’ videos are selected." What criteria define a video as "interesting"? Please provide details.
* L262 mentions "After rejecting videos for which our proposed automatic annotation method failed…" How were these videos rejected? Was it done manually or through an automatic process? More details are needed.
* Although the HowToGround dataset is generated from aggregated image-level annotations, this approach may struggle with capturing interactions that require understanding the temporal dimension, such as determining whether someone is walking forwards or backwards. Is this a limitation of your dataset and the resulting model? Quantitative evaluation of actions requiring temporal interactions could strengthen the paper.
* How does the VideoGLaMM generalize to other datasets, apart from the HowToCaption dataset?

---

### Official Review · Reviewer_NEaU · 2024-11-03

**Soundness:** 2
**Presentation:** 2
**Contribution:** 2
**Rating:** 3
**Confidence:** 5

**Summary:**

This submission gives a task, dataset, and model for grounded video caption generation. The main contributions are:

* They define the task as GROunded Video Caption Generation (GROC) and create a manually annotated test dataset specifically for this task.

* They design an automatic annotation pipeline that uses an existing model for grounding and a LLM for frame-level and further video-level captioning. They use this approach to the randomly selected subset of HowTo100M, and generate the final HowToGround training dataset.

* They propose a VideoGLaMM model, trained on the collected HowTo100M dataset, which achieves best performance on grounded video caption generation.

**Strengths:**

1. The released training and evaluation datasets could be useful in the field of video captioning after addressing the necessary ethics concerns.

**Weaknesses:**

I would say the most important weakness of this paper is **over-claim, fails to discuss and compare previous published research and show no respect to previous work**. Let's start with the beginning:

1. The authors claim that they propose a new task called grounded video caption generation (GROC). However, I don’t think so at all. Please refer to paper [1] titled "Grounded Video Description." I think that, in vision-and-language research, most researchers use the terms "video description" and "video captioning" interchangeably (the slight difference may be that "video description" sometimes includes more details). Paper [1] is the first to collect grounded video-text datasets and propose a model that uses grounding information to generate better video descriptions. They didn’t even claim that this task is a so-called new task because it is viewed as using alternative guidance to improve video description generation, not as a new task. One quick and intuitive way to compare these two is to check Figure 1 in this submission and Figure 2 in paper [1]. Thus, **I completely reject the claim that this task is newly proposed by the authors of this submission**.

2. The more interesting thing is that the authors actually cited paper [1] but **did NOT mention it at all in the main pages**. Instead, they "secretly" placed it in the Appendix. In my opinion, a paper with so much overlap should be carefully discussed in the related work section, in section 3 where the authors claim their novelty, and also at the end of section 5, where the authors describe how they construct their datasets.

3. As for the experiments, the proposed model is also lacking of novelty. Put more relevant papers here, like [2, 3, 4], which are also not properly cited and discussed in this submission. Thus, **I also clearly reject the claim in line 055-056 that "At the same time, producing natural language video descriptions where the described objects are spatio-temporally grounded with bounding boxes in videos has received much less attention"**. From my understanding, the authors replace the previsous LSTM networks with large language models. For sure, LLMs can get better results than LSTMs. However, these methods and models are not compared in their experiments. I don't believe that, in the era of LLMs, simply replacing the language module in previous methods with an LLM can be considered a big innovation in the overall model structure.

In summary, I clearly think this submission needs major revisions and a complete reframing in order to meet the publication standards of a conference like ICLR.

References:

[1] Grounded Video Description, Zhou et al. CVPR 2019.

[2] Learning to Generate Grounded Visual Captions without Localization Supervision, Ma et al. ECCV 2020.

[3] Attend and Interact: Higher-Order Object Interactions for Video Understanding, Ma et al, CVPR 2018.

[4] Comprehensive Visual Grounding for Video Description, Jiang et al. AAAI 2024.

**Questions:**

Most of my major concerns can be seen in the weakness section. Here I leave a few more:
1. Line 229-230 Sec. ?? -> Reference is missing.
2. The baselines are insufficient. One simple strategy is to directly feed the frames with grounding boxes into a video LLM. This is a widely used method called visual prompting, which can be tested in this training dataset without any additional structure or model tuning.
3. The ablation study is not convincing to me. As shown in Figure 3, I think most of the audience would like to know whether including grounded information really helps with video captioning; that is, a comparison of removing and keeping the grounding video encoder.

**Details Of Ethics Concerns:**

1. The GROC evaluation dataset is labeled by human annotators, and the authors only mention their protocols for human annotators in Appendix F. The authors may need to disclose more details regarding the labeling template, the platform they use, the salary of annotators, and other factors to ensure that the labeling process is a fully responsible practice.

2. The collected and planned-to-be-released training dataset is generated fully automatically with the help of LLMs. Safety and bias issues need to be addressed during the generation process, which are ignored in this paper.

---

### Official Review · Reviewer_PcHZ · 2024-11-05

**Soundness:** 2
**Presentation:** 3
**Contribution:** 3
**Rating:** 6
**Confidence:** 4

**Summary:**

The paper proposed a new task (GROC), a manually annotated test dataset, and an automatically generated dataset. It introduces VideoGLaMM, a model designed to generate captions and track object bounding boxes over time. Key features include spatio-temporal adapters, a bounding box decoder, and a temporal objectness head, all aimed at bridging multimodal understanding for video and language.

**Strengths:**

Novel Task, Dataset, and Model: The paper introduces a new task with a manually annotated evaluation set and a large-scale automatically generated dataset.
Dataset build up: Uses LLMs to generate consistent video-level captions from frame-level annotations, improving temporal consistency.
Model that support vlm to output bbox for video: VideoGLaMM incorporates spatio-temporal adapters and temporal objectness, supporting consistent tracking of objects in video.

**Weaknesses:**

Limited Model Comparison: It would be better for VideoGLaMM to include comparisons against additional models like InternVL and Qwen-VL, which also produce bounding boxes in text format, to strengthen its evaluation. Since the paper aims to establish a new task, a more comprehensive benchmark would enhance the reliability of its proposed contributions.

Discussion Scope: Additional discussion is needed on alternative methods for VLM bounding box outputs, specifically addressing why GLaMM was prioritized over other methods like [1,2]. This would help clarify its selection rationale within the broader context of existing approaches.

[1] Bannur, Shruthi, et al. "MAIRA-2: Grounded Radiology Report Generation." arXiv preprint arXiv:2406.04449 (2024).
[2] Zou, Ke, et al. "MedRG: Medical Report Grounding with Multi-modal Large Language Model." arXiv preprint arXiv:2404.06798 (2024).

**Questions:**

Clarification on Novelty: Needs a clearer explanation of differences from previous work, specifically the distinct contributions beyond those in “Grounded Video Description” (Zhou et al., 2019).

---

### Note · Authors · 2024-11-13

I have read and agree with the venue's withdrawal policy on behalf of myself and my co-authors.